# Altered SERCA Expression in Breast Cancer

**DOI:** 10.3390/medicina57101074

**Published:** 2021-10-08

**Authors:** Panayiota Christodoulou, Andreas Yiallouris, Artemis Michail, Maria-Ioanna Christodoulou, Panagiotis K. Politis, Ioannis Patrikios

**Affiliations:** 1School of Medicine, European University Cyprus, Nicosia 2404, Cyprus; Pa.christodoulou@euc.ac.cy (P.C.); i.patrikios@euc.ac.cy (I.P.); 2Center for Basic Research, Biomedical Research Foundation of the Academy of Athens, 11527 Athens, Greece; artemisbio@gmail.com (A.M.); ppolitis@bioacademy.gr (P.K.P.); 3Tumor Immunology and Biomarkers Laboratory, Basic and Translational Cancer Research Center, Department of Life Sciences, European University Cyprus, Nicosia 2404, Cyprus; Mar.Christodoulou@euc.ac.cy

**Keywords:** SERCA pump, SERCA genes, breast cancer, SERCA alterations

## Abstract

*Background and Objectives:* Calcium (Ca^2+^) signaling is critical for the normal functioning of various cellular activities. However, abnormal changes in cellular Ca^2+^ can contribute to pathological conditions, including various types of cancer. The maintenance of intracellular Ca^2+^ levels is achieved through tightly regulated processes that help maintain Ca^2+^ homeostasis. Several types of regulatory proteins are involved in controlling intracellular Ca^2+^ levels, including the sarco/endoplasmic reticulum (SR/ER) Ca^2+^ ATPase pump (SERCA), which maintains Ca^2+^ levels released from the SR/ER. In total, three ATPase SR/ER Ca^2+^-transporting (ATP2A) 1-3 genes exist, which encode for several isoforms whose expression profiles are tissue-specific. Recently, it has become clear that abnormal SERCA expression and activity are associated with various types of cancer, including breast cancer. Breast carcinomas represent 40% of all cancer types that affect women, with a wide variety of pathological and clinical conditions. *Materials and methods*: Using cBioPortal breast cancer patient data, Kaplan–Meier plots demonstrated that high ATP2A1 and ATP2A3 expression was associated with reduced patient survival. *Results:* The present study found significantly different SERCA specific-type expressions in a series of breast cancer cell lines. Moreover, bioinformatics analysis indicated that ATP2A1 and ATP2A3 expression was highly altered in patients with breast cancer. *Conclusion*: Overall, the present data suggest that SERCA gene-specific expressioncan possibly be considered as a crucial target for the control of breast cancer development and progression.

## 1. Introduction

Intracellular calcium ion (Ca^2+^) signaling [1] pathways are involved in key cellular processes, including excitation–contraction coupling, stimulus–secretion coupling, gene expression, control of the cell cycle, cell motility, autophagy, and apoptosis [2,3,4]. The cytosolic Ca^2+^ concentration is tightly maintained (~10^−7^ mol/L), with small amounts being released from several organelles (~10^−5^ mol/L) or through influx from the extracellular reservoir (~10^−3^ mol/L), that can generate marked signals to activate downstream signaling cascades [5]. However, prolonged intracellular Ca^2+^ elevation can be harmful and induces cell death [6]. Alteration/deregulation of Ca^2+^ signaling is a feature of numerous types of diseases, including cancer, heart failure, and neurodegenerative diseases [1]. The sarco/endoplasmic reticulum (SR/ER) Ca^2+^ ATPase pump (SERCA) belongs to the P-type pump family, which includes a large number of evolutionarily related Ca^2+^ pumps. There are three SERCA ATPase SR/ER Ca^2+^-transporting (ATP2A)1-3 eukaryotic genes that exist, which encode for 14 isoforms with tissue-specific expression profiles [7,8,9,10]. All of the isoforms have 75% homology and are cell-type-dependent [11]. *SERCA1* has two transcripts, 1a and 1b, which are present in adult and fetal skeletal muscle, respectively [12]. SERCA 2 consists of two transcripts, 2a, which is mostly found in cardiac and adult skeletal muscle, and 2b, which is abundant in both adult and fetal cardiac and skeletal muscles as well as in non-muscle cells. SERCA3 can be transcribed to three isoforms, SERCA3a–3c, all expressed in non-muscle cells [13]. All SERCA isoforms have been reported to be key players in cancer progression, proliferation, and survival [13,14]. Several studies have reported alterations in SERCA gene-specific expression in various types of cancer, suggesting that levels of SERCA may be potential therapeutic targets and biomarkers in cancer [1].

Cancer development and progression towards metastasis are known to be associated with alterations in post-translational modifications, gene copy numbers, epigenetics, and metabolic changes [15,16,17,18]. These changes are frequently observed as a result of alterations of the Ca^2+^ flux across the plasma membrane or across intracellular organelles [1].

Alterations in the structure of the ATP2A2 gene have been reported in colon and lung cancer. Specifically, 13 different novel alterations of the ATP2A2 gene have been reported in 27 of 416 alleles in patients, including intronic deletions, single-nucleotide alterations, missense mutations, and intronic insertions. Conversely, reduced ATP2A2 expression levels due to alterations in gene-promoter activity have been found in patient tissue biopsies and may be an early event in tumorigenesis [7]. Additionally, ATP2A2 expression levels are significantly correlated with tumor grade and survival rates, where high ATP2A2 expression has been detected in patients with glioblastoma [9]. In a separate study, ATP2A3 gene downregulation has been reported in gastric and colon cancer types. In contrast, ATP2A3 overexpression reduces cell viability by promoting apoptosis in breast cancer cell lines [19]. 

A known SERCA inhibitor originally extracted from the plant *Thapsia garganica* is thapsigargin, which is known to induce cell death in human hepatoma cells [19,20,21,22]. Thapsigargin has also been reported to reduce cell proliferation and induce apoptosis through a caspase-dependent pathway [23]. Moreover, a modified version of thapsigargin, mipsagarin, is undergoing clinical trials and is considered to be a strong candidate for the treatment of hepatocellular carcinoma [24]. Another study in breast cancer indicated the involvement of SERCA3 in early stage lobular dysplasia, with low expression levels at more advanced stages of lobular tumorigenesis [25]. In invasive breast carcinoma, SERCA3 expression levels are significantly decreased compared with that in normal patients, indicating that SERCA3 expression is inversely associated with tumor differentiation and the degree of aggressiveness/malignancy of ductal carcinoma [18,24,25,26,27,28]. 

There is a continued need to identify novel targeted therapies for the treatment of breast cancer, in combination with low toxicity chemotherapy, especially for chemoresistant triple-negative breast cancer (TNBC). This paper investigates the role and significance of different calcium-dependent SERCA ATPase proteins in breast cancer development. 

## 2. Materials and Methods

### 2.1. Cell Culture and Reagents

MCF10A (Michigan Cancer Foundation-10 human breast epithelial cells), MCF7 (Michigan Cancer Foundation-7), MDA231 (M.D. Anderson Metastasis Breast Cancer, Houston, TX, USA”), T47D (mammary gland; derived from metastatic site: pleural effusion) cells were provided by Barbara Ann Karmanos Cancer Institute (Detroit, MI, USA) and cultured in Dulbecco’s medium supplemented with 10% fetal bovine serum and 1% penicillin/streptomycin (Gibco, Invitrogen, Waltham, MA, USA). Cells were grown in a humidified atmosphere at 37 °C containing 5% CO_2_ and were routinely tested for mycoplasma contamination.

### 2.2. Western Blotting

Total protein was isolated from cultured MCF10A, MCF7, MDA-MB-231, and T47D cells with lysis buffer RIPA plus protease inhibitor cocktail. The homogenates were centrifuged at 17.000× *g* for 15 min at 4 °C. The supernatants were collected, and protein concentration was measured with Bradford protein assay (Bio-Rad protein assay, Hercules, CA, USA). A total of 30 μg of protein samples was loaded each time into SDS-PAGE gels and transferred to nitrocellulose membranes (Whatman, Maidstone, UK) using the semi-dry transfer system (Bio-Rad). The membranes were blocked with 5% dry milk dissolved in Tris-buffered saline (1×) containing 0.1% Tween-20 for 1 h at room temperature (RT). The membranes were incubated with primary antibodies at 4 °C overnight, followed by secondary antibodies for 1.30 h at RT. The primary antibodies in the Western blot were mouse anti-Serca 1 (Abcam, Cambridge, UK, ab-2819), mouse anti-Serca 2 (Abcam, ab-2861), rabbit anti-Serca 3 (alomone labs, ACP-014), and mouse anti-beta actin (Sigma, Kawasaki, Kanagawa, A5441). The secondary antibodies were rabbit anti-mouse IgG (Sigma, A9044) (1:20.000 dilution) and goat anti-rabbit IgG (Sigma, A6154) (1:10.000 dilution). The protein bands were detected using the Western Luminescent Detection Kit. Western blot images were analyzed using ChemiDoc XRS system (BioRad) with actin used as the protein control, as previously described.

### 2.3. Cancer OMICS Data Analysis 

UALCAN is a web-based resource platform using pancancer gene expression analysis from TCGA OMICS data. Tumor Subgroup Gene Expression and Survival Analyses were downloaded from UALCAN [29]. The expression of ATP2A1, ATP2A2, and ATP2A3 was analyzed in order to explore their expressions in breast cancer. 

### 2.4. Data Analysis from Patients with Breast Cancer from cBioPortal

For TCGA pancancer atlas, expression data *n* = 1084 samples (publicly available cases) of breast cancer downloaded from the cBioPortal website https://www.cbioportal.org, accessed on 7 May 2021 in the form of *z*-score-transformed data were used.

### 2.5. Statistical Analysis

Statistical analysis was performed using ANOVA followed by a Dunnett’s post hoc test. *p* < 0.05 considered statistically significant.

## 3. Results

To determine the expression levels of SERCA-A1, A2, and A3 in breast cancer cells, the present study examined the protein expression profiles in MCF-7, MDA 231, and T47D cells and compared them to a non-transformed mammary breast cell line, MCF-10A. As Figure 1 demonstrates, SERCA-A1 expression was significantly higher in the MCF10A cells vs. the MDA 231 and T47D cell lines. SERCA-A2 was also significantly higher in the MCF-7, T47D, and MDA231 cells vs. the MCF10A cells. In contrast, SERCA-A3 expression levels were reduced in the MCF-7 and T47D cells. These findings suggested a differential expression profile of SERCA-A1-3 in various subtypes of breast cancer cell lines.

Subsequently, for the present study, we performed bioinformatics analysis using the cBioPortal database of clinical samples. As indicated in Figure 2A,B SERCA-A1-3 expression levels were elevated and altered in patients with breast cancer, where further analysis (Figure 3A–C) showed that SERCA-A1, A2, and A3 had a differential expression profile in breast cancer subclasses, including luminal, HER2-positive, and TNBC. As such, ATP2A1 was found to be expressed at slightly higher levels in luminal, HER2-positive, and TNBC types vs. normal samples. ATP2A2 expression levels were not altered in any of the subclasses of breast cancer vs. normal samples. Interestingly, ATP2A3 expression levels exhibited a trend of increased levels in the luminal and HER2-positive samples vs. the normal controls and marginally increased in TNBC vs. normal samples (Figure 3A–C). Moreover, SERCA-A1, A2, and A3 differential expressions were observed in individual patients with colorectal adenocarcinoma, liver hepatocellular carcinoma, and prostate adenocarcinoma (Figure 4). 

Finally, a strong association between the specific expression profiles of the various SERCA genes and the survival rates of patients with different breast cancer subclasses was observed, as shown by Kaplan–Meier plots (Figure 2A). These data indicated that the reduced survival rates were associated with differential expression of the SERCA pump subunits. The high expression level of ATP2A3 was associated with a significant reduction in the monthly survival rates. Similarly, the ATP1A1 lever was elevated and affected breast cancer patients’ survival, whereas ATP2A2 seemed to be neutral in respect to survival.

Potentially, the present results could be considered as indicative of a crucial role of SERCA in breast cancer and thus a possible target for a new generation of treatments.

## 4. Discussion

The present study investigated the association between altered SERCA-specific gene expression levels in various breast cancer subtypes. 

Analyzing TCGA and RNA sequencing data of patients with breast cancer showed that SERCA-A1 and SERCA-A3 levels were increased in tumor samples vs. normal samples. Additionally, this SERCA-specific gene expression was correlated with the malignancy and prognosis of the breast cancer cell lines. These results supported the potential role of SERCA in the development of breast cancer. However, in order to reveal the soundness of these data, those should be validated in vitro on cell lines originated from each subtype and also in large cohorts of patients.

Several previous studies support the findings of the present study, including the link between ATP2B2 and ATP2A3 in cancer, as well as the hypothesis that ATP2B3 might serve as a possible cancer biomarker [7,18,29,30].

Intracellular Ca^2+^ signaling plays a key role in mediating pathways such as cell proliferation, differentiation, and survival. Ca^2+^-dependent release of Ca^2+^ from the ER that is mediated by secondary messengers, such as inositol-1,4,5-*tris*-phosphate, depends entirely on SERCA activity [31,32,33,34]. Therefore, SERCA-dependent Ca^2+^ transport and SERCA activity constitute a major negative feedback mechanism in the mobilization of Ca^2+^ within the cell. As such, SERCA-regulated Ca^2+^-concentration-dependent pathways are a key component to cellular Ca^2+^-related signal transduction pathways, that if altered, can lead to a cascade of events that can affect various cellular processes, including cellular proliferation.

A number of experimental studies have shown that changes in Ca^2+^ levels play an important role in breast cancer prevention and breast cancer cell proliferation [35]. Moreover, SERCA activity and Ca^2+^ uptake are critical for the development of the differentiated breast acinar phenotype, and defects in Ca^2+^ accumulation in the ER are associated with altered SERCA expression. These alterations may be involved in the early steps of breast cancer tumorigenesis [36]. Furthermore, the finding that altered SERCA3 expression is associated with several histological and molecular markers of ductal carcinogenesis indicates that altered Ca^2+^ levels are associated with remodeling during tumorigenesis in the breast epithelium [37]. 

TNBCs are the most malignant form of breast cancer, and numerous pharmacological trials on patients with TNBC have failed to yield beneficial results. Moreover, the molecular changes underlying the switch between the different breast cancer subtypes towards TNBC and their metastasis need to be further characterized. An improved understanding could possibly lead to novel combinatorial therapies that could reduce the high and sometimes fatal toxicity of chemotherapeutic drugs.

Altered SERCA isoform expression and SERCA-specific gene mutations have been documented in various types of cancer, including lung, colon, and leukemia [7,38,39,40,41,42]. Progress in SERCA-specific isoform activities and their role in Ca^2+^ signaling will enhance the understanding of how dysregulation in SERCA activity plays a role in tumorigenesis. The present findings reinforce this knowledge in relation to the involvement of altered SERCA-specific gene expression in breast cancer and highlight the need to identify specific and selective pharmacological SERCA isoform inhibitors. SERCA isoforms are differentially expressed in cancer *vs*. normal samples. ATP2A1 expression negatively affected patient survival rates, especially in luminal, HER2-positive cancer and TNBC vs. normal controls, whereas ATP2A2 levels remained unaltered. ATP2A3 expression was significantly increased in luminal and HER2-positive vs. normal samples. Overall, high expression levels were associated with a reduction in the survival rate. However, further clinical studies on a larger sample of cancer patients are required to evaluate the present results. The type of human sample used as well as the sample size are considered as the limitations of the present study.

## 5. Conclusions

Concluding, our findings suggested that SERCA pump and altered Ca^2+^ signaling, as well as dysregulation of Ca^2+^ homeostasis, had an effective relation, contributing to breast cancer development and progression. A potential future challenge will involve the development of novel therapeutic approaches targeting specific SERCA-related malfunctions with the aim of limiting cancer development and progression.

## Figures and Tables

**Figure 1 medicina-57-01074-f001:**
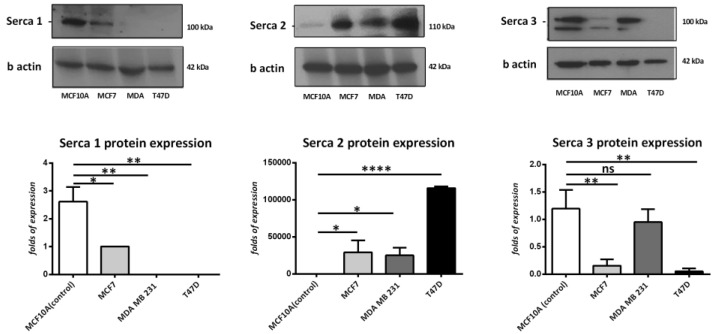
Western blot analysis of SERCA isoforms in breast cancer cell lines (MDA-231, MCF-7, MCF10A, and T47D). The analysis was performed in three independent experiments (*n* = 3), except in the case of Serca1, where *n* = 2. * *p* < 0.05, ** *p* < 0.01, **** *p* < 0.0001.

**Figure 2 medicina-57-01074-f002:**
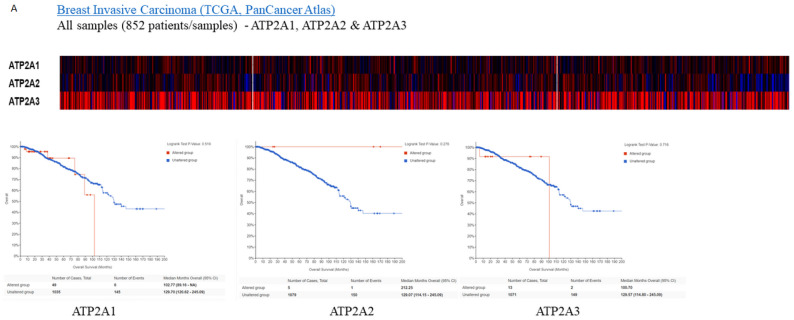
Expressions heatmap of SERCA isoforms in breast cancer patients. Kaplan–Maier plots. (**A**) ATP2A1 KM survival plots with expressions and alterations. *p* = 0.149. (**B**) ATP2A2 KM survival plots with expressions and alterations. *p* = 0.455.

**Figure 3 medicina-57-01074-f003:**
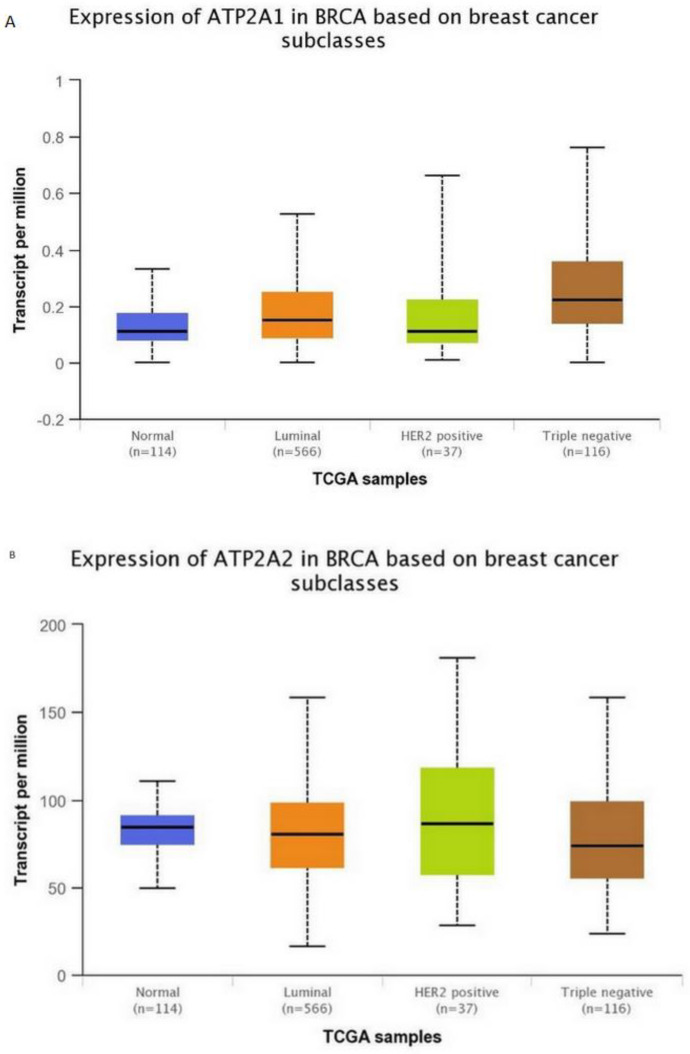
SERCA isoforms expressions. (**A**) ATP2A1 expression in breast cancer subclasses. (**B**) ATP2A2 expression in breast cancer subclasses. (**C**) ATP2A3 expression in breast cancer subclasses. Normal *n* = 114, luminal *n* = 566, HER-positive *n* = 37, and triple-negative *n* = 116. Data collected from: http://ualcan.path.uab.edu/.

**Figure 4 medicina-57-01074-f004:**
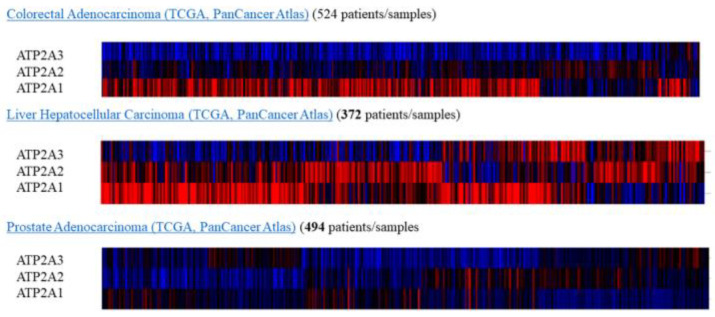
Expressions heatmap of SERCA isoforms in colorectal adenocarcinoma, liver hepatocellular carcinoma, and prostate adenocarcinoma. All data (TCGA PanCancer Atlas) collected from https://www.cbioportal.org/.

## Data Availability

The data presented in this study are openly available in https://www.cbioportal.org/ and http://ualcan.path.uab.edu/.

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
