# Peer review of "Altered SERCA Expression in Breast Cancer"

_medicina, 2021, doi:10.3390/medicina57101074_

Round 1

Reviewer 1 Report

Review of the manuscript medicina-1375617 titled “Altered SERCA Expression in Breast Cancer”

In this article, Panayiota et al. characterize the expression levels of three calcium ATPase pump genes SERCA1-3 in breast cancer cell lines and in publicly available patient sample data. The authors claim that there are different expression profiles for each SERCA and their expression levels different between the subtypes of breast cancer.  The authors also claim that SERCA expression is associated with patient survival and therefore SERCA is a potential target for cancer treatment. While I commend the authors for incorporating patient data and experimental data for their analysis, I am not convinced that their claims are fully supported by the evidence they present. The authors fail to present any functional role for SERCA genes in breast cancer and rule out the possibility that alterations in SERCA expression levels are just passenger events. More importantly, none of the patient data analysis seem to reach statistical significance levels as they are depicted.

Major concerns:

  1. SERCA expression might just be a downstream passenger event to an actual driver event that takes place in breast cancers. Associations between gene expression and survival do not always indicate a functional role for a that gene. Therefore, if the authors want to make the claim that SERCA is an important therapeutic target, then they should provide functional assays using cell lines where the SERCA expression levels are manipulated by si-/sh-RNA or CRISPR technologies. They should at least check the effect of SERCA knockdown on proliferation and cell cycle (and potentially on migration and invasion). Alternatively, the authors can use the inhibitors that they mentioned, thapsigargin or mipsagarin, to show any effect of reducing SERCA activity in cancer cells.
  2. The authors claim that there are differences in SERCA expression among different subtypes of breast cancer. However, this does not seem to be the case in the patient data in figure 3. None of the p-values indicated are below a significant threshold of 0.05. I also went on https://ualcan.path.uab.edu/ where the authors have performed their analyses. Even pairwise comparisons between different subgroups do not yield any significant results. Since they cannot support their hypothesis with the patient data, the authors should consider selecting at least 2 cell lines that belong to each subtype of breast cancer and support their argument experimentally. Their western blots in Figure 1 only include 1 normal cell line (MCF10A), 2 luminal A cell lines (MCF7, T47D), and only 1 basal cell line (MDA-MB-231).

Similarly, the p-values indicated for the Kaplan-Meier curves are not significant either, which suggests that there is no significant association between SERCA gene expression and survival. This is opposite to what the authors are claiming.

Minor concerns:

  1. The authors use “TNBC” and “basal” subtype interchangeably, but they should acknowledge that not all TNBCs are basal, or vice versa.
  2. Figure 1 shows quantification of SERCA1-3 expression from three independent experiments (ExpA-C). However, “a-Serca1.TIF” blot image shows a lot of background in general and especially a large smudge right where ExpC samples are. How did the authors manage to quantify the 3rd experimental results from these blots? Individual data points should be overlayed with the bar plots depicted in Figure 1. If ExpC was not quantified for Serca1 expression, then this experiment should be repeated to maintain at least n=3 independent experiments.

Editorial comments:

  1. Figure legends have subpanels such as A,B,C etc. however the figures themselves are not labeled for these subpanels.
  2. The authors should indicate how the patients (columns) within the heatmaps in Figure 2 and 4 are ordered. cBioportal usually orders the patients based on the mutation oncoplot, and not gene expression.
  3. Lines 182 and 183 must be typos since they are not full sentences.
  4. In line 96, the authors should indicate what they used ivermectin for
  5. The manuscript could benefit from a grammar and spelling check.

Author Response

27th  September 2021

Dear Editor,

On behalf of my co-authors, I would like to thank you and the reviewers for the thoughtful comments on our manuscript entitled Altered SERCA Expression in Breast Cancer by Christodoulou Panayiota, Yiallouris Andreas, Michail Artemis, Christodoulou Maria-Ioanna, Politis K. Panagiotis and Patrikios Ioannis submitted for publication as an original research article in the “Medicina MDPI, Submitted to section: Oncology, special issue: “Cancer Treatment: Clinical Applications of Cell Cultures”.

We deeply appreciate the opportunity to revise our article after addressing reviewer’s comments, and we believe the suggested changes have added significant value to our manuscript.

The revised manuscript is now submitted in the Manuscript Tracking System of the Journal for re-evaluation. Below please find our responses to each of the Reviewer’s comments.

We hope that the issues raised by the reviewer have been adequately addressed.

Thank you for considering our revised manuscript.

Editorial comments:

  1. Figure legends have subpanels such as A, B, C etc. however the figures themselves are not labelled for these subpanels. (figure panels revised and corrected)

  1. The authors should indicate how the patients (columns) within the heatmaps in Figure 2 and 4 are ordered. cBioportal usually orders the patients based on the mutation oncoplot, and not gene expression. (The heatmap ordered using mRNA expression z-scores relative to normal samples (log RNA Seq V2 RSEM and sorted by case id alphabetically, Figure revised)

  1. Lines 182 and 183 must be typos since they are not full sentences. (Corrected)

  1. In line 96, the authors should indicate what they used ivermectin for (revised)

  1. The manuscript could benefit from a grammar and spelling check. (The manuscript has been revised by native English speaker)

Reviewer 1:

Major Comments #1

We thank the reviewer for the comment and we are in line with it. However the importance of SERCA as therapeutic target is well studied and proved including the death promoting effects of various SERCA pump inhibitors.

 https://pubmed.ncbi.nlm.nih.gov/15662118/

https://jhoonline.biomedcentral.com/articles/10.1186/s13045-020-01015-9

https://faseb.onlinelibrary.wiley.com/doi/abs/10.1096/fasebj.30.1_supplement.768.4

https://pubmed.ncbi.nlm.nih.gov/28972171/

https://www.nature.com/articles/srep35196

In addition, for the time given, is not feasible to order and perform si-/sh-RNA or CRISPR experiments. This is of course in our future plans.

Major Comments #2

We thank the reviewer for this comment. Indeed, there is no significant differences among the various subtypes. However, there is a trend of higher levels in the luminal and HER2 positive samples, only in the case of ATP2A3. Thus, the corresponding sentence in the Results sections (lines 143-145) has changed to: “Interestingly, ATP2A3 expression levels exhibited a trend of increased levels in the luminal and HER2 positive samples Vs the normal controls and marginally increased in TNBC Vs normal samples.” Similarly in the Discussion section the sentence in lines (240-241) has changed to: “ATP2A3 expression was slightly increased in luminal and HER2 positive Vs normal samples”.

We also agree with the fact that data from bioinformatics tools should be experimentally validated. However, in the time given is not feasible to include experiments from two cell lines originated from the four types analyzed; but this is among our near future studies. Based on that, we make this comment in the Discussion section (lines 188-190): “However, in order to reveal the soundness of these data, those should be validated in-vitro on cell lines originated from each subtype and also in large cohorts of patients

Minor Comments #1

We thank the reviewer for this comment, and we agree the two terms are not identical. The term “basal” is not included in the updated version of the manuscript. So, we only refer to the “TNBC” subtype.

Minor Comments #2

We would also like to thank the reviewer for pinpointing this discrepancy. We have now taken out from the measurement of the intensity of bands this specific western blot for serca 1. Thus, we have recalculated the fold of expression and corresponding p values for serca1 with n = 2. We have incorporated these changes in the revised Figure 1. We have also indicated this difference in the n value in the legend of revised Figure 1.

Reviewer 2:

 14 - "full stop" at the end of the sentence is missing. (Corrected)

182-183 sentences are unnecessary, meaningless. (Revised)

186 add "line" - cell "line" types observed. (Added)

In the chapter "Cell Culture and Reagents", the authors describe 5 cell lines, while in the chapter "results" the authors describe only 4 cell lines. The described MCF12F cell line is not shown in the results. I think it should be removed from the publication.  (MCF12F cell line removed)

Yours sincerely,

Andreas Yiallouris

BSc, MSc, PhD

Lecturer, Medical Biochemistry, School of Medicine, European University Cyprus

Reviewer 2 Report

Very interesting topic regarding investigating the role and significance of different Calcium depended SERCA ATPase proteins in breast Cancer development.

114 - "full stop" at the end of the sentence is missing.
182-183 sentences are unnecessary, meaningless.
186 add "line" - cell "line" types observed.

In the chapter "Cell Culture and Reagents", the authors describe 5 cell lines, while in the chapter "results" the authors describe only 4  cell lines.

The described MCF12F cell line is not shown in the results. I think it should be removed from the publication. 

Author Response

(The authors gave the same response as above.)

Round 2

Reviewer 1 Report

The changes that the authors have made have improved the quality of this review.